# Synthesis of TiO_2_-ZnO n-n Heterojunction with Excellent Visible Light-Driven Photodegradation of Tetracycline

**DOI:** 10.3390/nano14221802

**Published:** 2024-11-11

**Authors:** Ying Zhang, Xinkang Bo, Tao Zhu, Wei Zhao, Yumin Cui, Jianguo Chang

**Affiliations:** Anhui Provincial Key Laboratory of Green Carbon Chemistry, School of Chemistry and Material Engineering, Fuyang Normal University, Fuyang 236037, China; 18856643672@163.com (X.B.); zt18726384674@outlook.com (T.Z.); 13023093135@163.com (W.Z.); cymlh@126.com (Y.C.)

**Keywords:** ZnO, TiO_2_, tetracycline removal, n-n heterojunctions, photocatalysis

## Abstract

Zinc oxide-based photocatalysts with non-toxicity and low cost are promising candidates for the degradation of tetracycline. Despite the great success achieved in constructing n-n-type ZnO-based heterojunctions for the degradation of tetracycline under full-spectrum conditions, it is still challenging to realize rapid and efficient degradation of tetracycline under visible light using n-n-type ZnO-based heterojunctions, as they are constrained by the quick recombination of electron–hole pairs in ZnO. Here, we report highly efficient and stable n-n-type ZnO-TiO_2_ heterojunctions under visible light conditions, with a degradation efficiency reaching 97% at 1 h under visible light, which is 1.2 times higher than that of pure zinc oxide, enabled by constructing an n-n-type heterojunction between ZnO and TiO_2_ to form a built-in electric field. The photocatalytic degradation mechanism of n-n TiO_2_-ZnO to tetracycline is also proposed in detail. The demonstration of efficient and stable heterojunction-type ZnO photocatalysts under visible light is an important step toward commercialization and opens up new opportunities beyond conventional ZnO technologies, such as composite ZnO catalysts.

## 1. Introduction

Antibiotic organic pollutants, such as oxytetracycline hydrochloride, tetracycline hydrochloride, doxycycline hydrochloride, doxycycline hydrochloride, minocycline hydrochloride, chlortetracycline hydrochloride, and metacycline hydrochloride, pose serious health and environmental risks due to their properties [1]. Various technologies, including adsorption [2], filtration [3,4], and biodegradation [5,6], have been utilized for antibiotic removal [7]. However, conventional methods like adsorption and filtration detach antibiotics without complete biodegradation. Therefore, novel technologies are necessary for the effective elimination of antibiotics in wastewater to mitigate their harmful effects on human health and the environment.

Photocatalytic oxidation technology, a widely used method, has recently captured our interest in treating antibiotics due to its excellent direct utilization of sunlight [8]. This method relies on developing a proficient photocatalyst with sufficient light-harvesting capability and efficient charge separation [9]. In recent studies, ZnO has been widely explored for its potential to remove antibiotics to purify water ecosystems, owing to its strong UV light-activated properties and non-toxic characteristics [10]. However, the photocatalytic performance of ZnO is limited by the high recombination rates of photogenerated electron–hole pairs and its narrow visible light absorption range [11]. Therefore, there is notable scientific interest in the development of a narrow-band-gap photocatalyst for the removal of antibiotics.

Combining zinc oxide with metal oxides to construct homotypic heterojunctions, such as TiO_2_/ZnO [12], Cu_2_O/ZnO [13], and WO_3_/ZnO [14], has been shown to significantly boost photocatalytic activity by promoting the rapid transfer of photoinduced charge carriers and increased surface hydroxyl groups. It is commonly accepted that the presence of surface hydroxyl groups on the catalyst can enhance the degradation rate of tetracycline [15]. Recent studies have also highlighted the efficacy of TiO_2_ as a catalyst in the photodegradation of tetracycline solution [16].

To address the issues associated with enhanced charge separation, our study aims to synthesize an n-n heterojunction between anatase TiO_2_ and ZnO through a hydrothermal method for tetracycline degradation. Achieving a 97% degradation rate in 1 h (1.2 times that of pure ZnO), we mitigate the electron–hole recombination issue inherent in ZnO and bolster its photocatalytic activity. The photocatalytic efficiency of anatase TiO_2_-modified ZnO will be evaluated under visible light illumination using tetracycline as a model antibiotic. Additionally, we propose to investigate the active species, charge transfer kinetics, and degradation mechanism that amplify photocatalytic activity by constructing heterogeneous structures and advancing ZnO-based technologies. This study aims to deepen our understanding of the mechanism behind antibiotic photodegradation.

## 2. Experimental Section

### 2.1. Chemicals

Tetrabutyl titanate (TBOT, purity 98%), ethanol (purity 99.7%), CTAB (purity 99%), urea (purity 99%), zinc acetate (purity 99%), and tetracyclines (purity 98%) were obtained from Sinopharm Corporation Ltd. (Beijing, China) All chemicals were utilized as received without additional purification.

### 2.2. Preparation of TiO_2_-ZnO (TZ)

A series of TiO_2_-ZnO composites with varying TiO_2_ spheres were synthesized using hollow TiO_2_ spheres as a starting material. Firstly, hollow TiO_2_ spheres were prepared according to previously published methods [7]. Secondly, a series of hollow TiO_2_ spheres (10 mg, 30 mg, 50 mg, 70 mg, 90 mg) was ultrasonically dispersed in 20 mL ethanol for 1 h, followed by the addition of specific amounts of CTAB, urea, and zinc acetate and continuous stirring for 5 h to form a suspension. The resulting mixture was transferred into Teflon-lined stainless-steel autoclaves and heated in a vacuum-drying oven at 140 °C for 4 h. The product was then washed thrice with ethanol and dried at 45 °C for 10 h, yielding the TiO_2_-ZnO (TZ) photocatalyst. Pure ZnO was prepared using the same method without the inclusion of a hollow TiO_2_ sphere.

### 2.3. Catalyst Characterization

SEM and TEM techniques were employed to characterize the morphology and structure of the TiO_2_-ZnO sphere; XRD was used to analyze the crystal structure of the TZ photocatalyst; XPS analysis was conducted to assess the chemical state of the TZ surface elements; and solid ultraviolet spectroscopy was utilized to characterize the photoelectric properties of TZ.

### 2.4. Photocatalytic Activity Tests

The photocatalytic performance of the synthesized materials was evaluated by degrading tetracycline (TC: 20 mg/L) under visible light (AM 1.5G) from a 300 W Xe lamp (Beijing Perfect Light Technology Co., Ltd., Beijing, China) with a cutoff filter. Typically, a 10 mg catalyst was dispersed in a 50 mL beaker containing TC solutions, sonicated for 2 min, and stirred for 20 min in darkness to reach the adsorption–desorption equilibrium. Subsequently, with continuous stirring, the Xe lamp was turned on to initiate the photocatalytic reaction. Samples (3.0 mL) were collected from the suspension every 30 min and centrifuged to obtain the supernatants for further analysis. The solution concentration was determined using a UV-vis spectrometer by monitoring the characteristic absorbance at 357 nm for TC [17].

## 3. Results and Discussion

### 3.1. Morphology Analysis

Figure 1 briefly depicts the preparation process of the TZ photocatalyst. Initially, CPS microspheres were prepared via emulsion polymerization. Subsequently, a layer of amorphous TiO_2_ was adsorbed onto the surface of the CPS template and calcined at 450 °C [18], as depicted in Appendix A. Then, anatase TiO_2_ was used as a modifier and combined with zinc acetate to form a precursor solution in pure ethanol solvent conditions. Finally, anatase TiO_2_ was loaded via a hydrothermal reaction and deposited onto the surface of wurtzite zinc oxide, resulting in the TiO_2_-ZnO photocatalyst.

To further understand the distribution characteristics of TiO_2_-ZnO and ZnO, TEM and high-resolution transmission electron microscopy (HRTEM) were employed to analyze TZ and ZnO, as depicted in Figure 2 and Appendix A. Notably, several TiO_2_ nanoparticles were observed adhering to ZnO, with the crystal lattice spacings of TZ measuring 0.248 nm and 0.351 nm (Figure 2b), corresponding to the (100) crystal plane of ZnO [19] and the (101) crystal plane of TiO_2_ [18]. Interestingly, this is consistent with the diffraction ring shown in Figure 2c, providing evidence for the formation of the TiO_2_-ZnO n-n heterojunction [20,21]. The Energy Dispersive X-ray spectroscopy (EDS) spectrum and elemental mapping images of TZ shown in Figure 2e–h demonstrate the heterojunction of TiO_2_/ZnO consisting of C, O, Ti, and Zn elements, proving that the elements are uniformly distributed over TZ. The presence of the n-n heterojunction between TiO_2_ and ZnO creates a built-in electric field that facilitates electron transport [22]; electrons will migrate directionally from ZnO to TiO_2_.

### 3.2. Phase Analyses

As shown in Figure 3, one of the TZ samples presents stronger characteristic peaks located at 31.66(100), 34.34(002), 36.12(101), 47.40(102), 56.42(110), 62.76(103), 66.16(200), 67.84(112), 69.00(201), 72.48(004), and 76.88(202), which is consistent with the wurtzite ZnO results (JCPDS:36-1451) [23]. Additionally, the diffraction peaks of TiO_2_ located at 25.14(101), 37.91(004), 48.38(200), and 55.21(211) have been observed in TZ, which is consistent with the diffraction peaks of TiO_2_ (JCPDS: 21-1272) [24]; these correspond to the (101), (004), (200), and (211) crystal planes of anatase TiO_2_, which means that the prepared TiO_2_-ZnO was based on anatase TiO_2_ nanoparticles. The XRD results notably demonstrate the presence of anatase TiO_2_ and wurtzite ZnO crystal forms in TZ, which is advantageous for the photodegradation of TC.

### 3.3. Binding Environment Analysis

XPS analysis was performed to further understand the chemical composition of TiO_2_, ZnO, and TZ samples. The full spectrum of TZ (Appendix A) discloses the appearance of characteristic Ti2p, Zn2p, O1s, and C1s peaks clearly in Figure 4.

As shown in Figure 4a, the Ti2p XPS spectrum of TZ (Figure 4a) exhibits four peaks at 464.29 eV and 458.61 eV, referring to Ti 2p_1/2_ (~464 eV) and Ti 2p_3/2_ (~458 eV), respectively, which matched the characteristic features of TiO_2_ [25]. Notably, the Ti 2p XPS peak of TZ exhibited a notable negative shift, indicating a higher electron density of Ti atoms in the TZ samples than in TiO_2_. This suggests the formation of a strong interaction between ZnO and TiO_2_. Interestingly, the separation (5.68 eV) is observed between Ti 2p_1/2_ (464.29 eV) and Ti 2p_3/2_ (458.61 eV), ascertaining the existence of the Ti^3+^ state in TiO_2_ [26]. The existence of Ti^3+^ in the surface lattice of TiO_2_-ZnO was also associated with the creation of oxygen defects, which can have a significant impact on photocatalytic oxidation reactions [27].

Figure 4b shows two symmetric binding energy peaks of Zn 2p_3/2_ and Zn 2p_1/2_ at 1021.51 eV and 1044.49 eV, respectively, with a binding energy difference of 22.99 eV, confirming that Zn is in the +2 oxidation state [28]. In comparison to pure ZnO, the Zn 2p signals of TZ shifted to higher binding energies. Notably, these positive shifts suggest a significant interaction between ZnO and TiO_2_, illustrating electron transfer from ZnO to TiO_2_ upon their contact.

As shown in the O1s spectrum (Figure 4c), O1s could be split into two different peaks at 530.09 and 531.71 eV corresponding to lattice oxygen (Zn-O/Ti-O) and surface active oxygen vacancies (O_*V*_). Compared to ZnO and TiO_2_, TZ exhibits an increased shift in the O_*V*_ peaks to 531.71 eV. This shift is advantageous for the photoinduced charge transfer, resulting in a high catalytic activity [29].

Subsequently, the C1s peaks of TZ exhibited varying structures of carbon. Deconvolution of the C1s spectra of TZ (Figure 4d) revealed peaks at binding energies of 284.8 eV, attributed to sp^2^ C–C and sp^3^ C–C of inorganic carbon (diamond structure), 286.09 eV corresponding to C–O, and a broad peak at 288.86 eV indicative of C=O [30]. Based on the aforementioned results, it can be concluded that anatase TiO_2_-modified ZnO was successfully synthesized, consistent with the XRD findings.

### 3.4. Photocatalytic Activity

#### 3.4.1. Photocatalytic Degradation of TC Contaminant

The photocatalytic activity of TZ samples was assessed through the degradation of TC under visible light irradiation. ZnO and TiO_2_ were employed as reference photocatalysts to explore the impact of the anatase TiO_2_ modifier.

The introduction of modified ZnO was achieved through the reaction of TiO_2_ with Zn(Ac)_2_. As shown in Appendix A, this study investigated the effect of different dosages of TiO_2_ (10 mg, 30 mg, 50 mg, 70 mg, 90 mg) on the photocatalytic performance of TZ materials, with a constant amount of Zn(Ac)_2_. The optimum photocatalytic performance of the TZ catalyst was observed when the addition of TiO_2_ was 50 mg (Appendix A). Any increase or decrease in the amount of TiO_2_ resulted in a decline in photocatalytic efficiency (Appendix A).

As exhibited in Figure 5a, TC degradation hardly occurs without catalysts, indicating that the self-photolysis and stability of TC is insignificant [31]. The dark reaction was extended to 60 minutes, a state of equilibrium in desorption was still established (Appendix A). The catalytic performance of ZnO is satisfactory, with TC degradation efficiencies of 75% within 1 h, primarily due to the rapid recombination of photogenerated carriers. Remarkably, the photocatalytic efficiency is significantly enhanced in TiO_2_-ZnO (dosages of TiO_2_:50 mg), achieving a 97% degradation efficiency of TC within 1 h, indicating the beneficial interaction between anatase TiO_2_ and ZnO and the construction of n-n heterojunctions. As represented in Table 1, the photodegradation efficiency of TiO_2_/ZnO was compared to other published ZnO photocatalysts applied for the photodegradation of TC under visible light for 1 h. These results show that TiO_2_/ZnO with an n-n heterojunction structure and abundant oxygen vacancies as active sites is a favorable photocatalyst under visible light.

Ultimately, complete TC degradation was accomplished after 100 min of sunlight irradiation. The degradation kinetics of TC were simplified using a pseudo-first-order model $-ln\frac{C_{t}}{C_{o}}=\kappa t$, where $C_{O}$ and $C_{t}$ represent the TC concentration at time 0 and *t*, respectively, and *k* represents the kinetic constant [28]. As shown in Figure 5b, TiO_2_-ZnO exhibits the highest TC degradation rate constant (*k* = 0.0675 min^−1^), nearly 5.67 times higher than that of ZnO (0.0119 min^−1^). This proves that anatase TiO_2_-decorated ZnO enhances the efficient separation of electron–hole pairs, resulting in enhanced catalytic performance.

Cycling experiments were conducted to assess the stability of TiO_2_-ZnO photocatalytic degradation of TC (Figure 6). As expected, compared to pure ZnO (Figure 6b), the photocatalytic degradation activity of TiO_2_-ZnO (Figure 6a) remained consistent (99%), with no noticeable decline in TC degradation efficiency after four repeated experiments. This further confirms the excellent stability of TiO_2_-ZnO, demonstrating its recyclability for photocatalytic reactions. The XRD spectrum and SEM image of recycled TiO_2_-ZnO is shown in Appendix A, where the positions and intensity of the characteristic peaks and morphology remain unchanged after four experimental runs, further implying the exceptional stability of the TiO_2_-ZnO in TC photocatalytic degradation application.

#### 3.4.2. Impact of Environmental Factors on Photocatalytic Degradation

Given that natural water bodies contain various inorganic ions, it is crucial to explore the impact of co-inorganic ions (SO_4_^2−^, NO_3_^−^, Cl^−^, Na^+^, K^+^) on the degradation of TC, as illustrated in Figure 7a. Interestingly, the presence of K^+^ leads to a slight reduction in the catalytic activity of TZ. On the contrary, the introduction of Na^+^ hinders the catalytic activity of TZ, possibly because Na^+^ competes for reactive sites, thereby impeding TC degradation. Furthermore, SO_4_^2−^, NO_3_^−^, and Cl^−^ also hinder the catalytic efficiency of TZ.

As shown in Figure 7b, it is observed that a higher concentration of TC is inversely related to the catalytic activity of TZ. This phenomenon can be primarily attributed to the competitive adsorption of TC on the surface of TZ, leading to a decrease in catalytic efficiency.

It is widely recognized that TC molecules exhibit tendencies for protonation and deprotonation in the reaction solution. Therefore, we conducted a detailed investigation into the impact of pH value (3, 5, 7, 9) on the degradation of TC by introducing HCl and NaOH. As shown in Figure 7c, a noteworthy enhancement in the photocatalytic activity of TZ is observed at a pH value of 5 (97.62%), compared to the neutral pH of 7 (97.21%).

This improvement can be primarily attributed to the shift toward a more positively charged surface state of TZ under acidic conditions, facilitating increased adsorption between TC and TZ and thereby promoting the catalytic reaction. Conversely, a pH value of 3 leads to a decrease (79.34%) in catalytic activity due to excessive acidity diminishing the ionization degree of the reactants. Furthermore, raising the pH to 9 results in a significant decline (90.91%) in TC degradation efficiency, as the active sites on the TZ surface become occupied by hydroxide ions (${OH}^{-}$) under alkaline conditions.

To evaluate the potential applicability of TZ, a study was conducted on the photocatalytic degraded TC in three distinct types of water bodies, as shown in Figure 7d [10]. Comparing the photocatalytic performance of TZ in distilled water, it was observed that the efficiency of TZ in mineral water, lake water, and tap water was notably diminished during TC degradation. This reduced catalytic activity can be attributed to the presence of inorganic salts and insoluble organic matter in the water samples. Nevertheless, despite the challenges posed by the varying water compositions, TZ still demonstrates significant efficacy in the practical realm of TC degradation, thereby reinforcing its potential utility in wastewater treatment applications.

### 3.5. Potentail Photocatalytic Mechanism of TiO_2_-ZnO

#### 3.5.1. Active Species Determination

To demonstrate the catalytic mechanism involved in the degradation of TC by TiO_2_-ZnO, a series of scavengers [24], including benzoquinone (BQ) for ${\cdot O}_{2}^{-}$, isopropanol (IPA) for ·OH, and ammonium oxalate (AO) for $h^{+}$, was introduced into the photocatalytic reaction to trap free radicals. As depicted in Figure 8a, the outcomes reveal the inhibitory effects of BQ and IPA on the photocatalytic degradation of TC, while the removal of TC remains largely unaffected following the addition of AO, indicating that $h^{+}$ does not significantly contribute to TC degradation. This outcome underscores the predominant roles of ${\cdot O}_{2}^{-}$ and ·OH as the primary active species in the photocatalytic degradation of TC using the TiO_2_-ZnO catalyst.

The electron spin resonance (ESR) technique was utilized to identify the reactive oxygen species involved in the degradation of TC by the TZ photocatalyst using 5,5-Dimethyl-1-Pyrroline-N-Oxide (DMPO). In comparison to ZnO (Figure 8b,c), TiO_2_ and TZ exhibit a more pronounced signal of ${\cdot O}_{2}^{-}$ and ·OH, underscoring the significant roles that these species play in the photocatalytic reaction. This outcome verified that modifying ZnO with anatase TiO_2_ can enhance catalytic activity by introducing oxygen vacancies, aligning well with the XPS results.

The signals for ${\cdot O}_{2}^{-}$ and ·OH were notably intensified in ZnO (Appendix A), indicating that ZnO could be efficiently activated by the photogenerated electrons in the illumination process. Conversely, as the illumination time increased, the signal intensities of ${\cdot O}_{2}^{-}$ and ·OH in anatase TiO_2_ were significantly decreased (Appendix A). Based on the observations from Appendix A, we deduced that ${\cdot O}_{2}^{-}$ and ·OH served as the primary reactive species in TZ, contributing to its enhanced activity in TC degradation.

#### 3.5.2. Charge Transfer Kinetics

The capacity of TZ to effectively separate photoinduced carriers plays a pivotal role in determining its photocatalytic activity. To investigate the spatial separation performance of TZ, a variety of techniques were employed, such as transient photocurrent responses (TPR), electrochemical impedance spectroscopy (EIS) plots, and steady-state photoluminescence (PL) spectra. Pure TiO_2_ and ZnO were employed as control samples for comparison.

Initially, TPR spectra were obtained using a conventional three-electrode setup while visible light was alternately turned on and off (Figure 9a). It was noted that the photocurrent density exhibited a notable increase or decrease upon switching on or off the lamp, respectively, indicating the remarkable photosensitivity properties of the samples. Intriguingly, TZ demonstrated the most noteworthy photocurrent density (2.5 mA cm^−2^), which was approximately 1.39 and 2.08 times higher than that of TiO_2_ (1.8 mA cm^−2^) and ZnO (1.2 mA cm^−2^), respectively. This indicates that the modification of ZnO with anatase TiO_2_ facilitates the efficient separation of electron–hole pairs compared to pure TiO_2_ or ZnO.

The conductivity and charge transport capabilities of TZ were assessed through EIS. As shown in Figure 9b, TZ displayed the smallest semicircle diameter compared to TiO_2_ and ZnO, indicating that TZ exhibits superior efficiency in the separation of photogenerated electrons and holes.

Furthermore, PL spectra were utilized to analyze the efficiency of charge carrier separation for TZ (Figure 9c). It was observed that ZnO-TiO_2_ displayed a lower PL intensity compared to pure ZnO, suggesting that the recombination of photoelectron–hole pairs in anatase TiO_2_-modified ZnO was effectively suppressed. Notably, ZnO exhibited a strong emission peak, suggesting rapid electron–hole recombination with limited charge utilization. Conversely, the heterojunctions of anatase TiO_2_-modified ZnO exhibited significantly reduced PL intensity, indicating the highly effective separation of photogenerated charges. Moreover, anatase TiO_2_-modified ZnO exhibited a prominent emission peak at around 442 nm and 523 nm, attributed to cyan and green emission, respectively. Cyan emission originated from the oxygen vacancy (O*_V_*), while green emission originated from the interstitial oxygen (O*_i_*) [28].

To investigate oxygen defect species, photoluminescence (PL) spectra in the visible region were analyzed by deconvoluting them into two Gaussian curves, as shown in Figure 9d. Moreover, compared to pure ZnO (64%), the percentage of the cyan emission peak for TZ (59%) decreased significantly due to the formation of oxygen vacancy (O*_V_*) internal defects, which generate superoxide anion radicals and promote strong oxidation [32]. The interaction between an oxygen vacancy (V*_O_*) and O_2_ can be described as $V_{O}+O_{2}\to V_{O}^{+}+\cdot O_{2}^{-}$. Concurrently, the percentage of the green emission peak area for TZ remarkably increased to 41%, indicating favorable O_2_ adsorption and enhanced charge separation due to electron capture between the ZnO and ZnO interface. These results suggest that regulating oxygen defect contents using anatase TiO_2_ as a modifier can effectively enhance the photocatalytic degradation activity of TC.

#### 3.5.3. Band Structures

The UV–visible absorption spectrum was employed to analyze the optical absorption capacities of the synthesized samples (TiO_2_, ZnO, TZ), as illustrated in Figure 10a. From Figure 10, it is evident that TiO_2_-ZnO exhibits higher absorbance of visible light compared to ZnO, indicating that TZ possesses an enhanced photon capture ability. This conclusion is further supported by the results shown in Figure 10b: the band gap of ZnO, determined to be 3.2 eV (wurtzite ZnO, consistent with the XRD results), is reduced to 3.1 eV upon modification with anatase TiO_2_, attributed to the formation of oxygen vacancies. This modification notably enhances the visible light catalytic activity.

The Mott–Schottky plots were utilized to evaluate the flat band potential ($E_{fb}$) of the synthesized samples. The positive slopes observed in the Mott–Schottky curves (Figure 10c,d) confirm that both the prepared ZnO and anatase TiO_2_ are n-type semiconductors [33], with flat band potentials of −0.29 V and −0.18 V (vs. NHE, pH = 7), respectively. In n-type semiconductors, the flat band potential ($E_{fb}$) is closely associated with the lower edge of the conduction band (CB) and the Fermi level. Based on the aforementioned DRS results, the band gaps for ZnO and anatase TiO_2_ were determined as 3.2 eV and 3.15 eV, respectively. To calculate the valence band energy ($E_{VB}$), the relationship among the band gap ($E_{g}$), conductions band, and valence band energies can be exploited; that is, $E_{VB}=E_{g}+E_{CB}$ [33]. The conduction band (CB) and valence band (VB) energies for ZnO were confirmed as −0.29/2.91 eV, and for anatase TiO_2_ as −0.18/2.97 eV.

#### 3.5.4. Photocatalytic Mechanism

Based on the results presented in Figure 11, this study proposes a feasible mechanism for the photocatalytic degradation of TC by TiO_2_-ZnO under simulated visible light. Initially, when these materials are exposed to light, electrons within the valence bands of TiO_2_ and ZnO absorb energy and transition to the conduction bands, generating electron–hole pairs [34]. This phenomenon is facilitated by the existence of a band gap between the valence and conduction bands of TiO_2_ and ZnO, thereby imparting them with photocatalytic activity [35]. Subsequently, under visible illumination, electrons in ZnO are excited to higher energy levels and then transferred to the conduction band of TiO_2_ [23], leading to interactions with oxygen molecules, resulting in the creation of superoxide anion radicals. The TiO_2_ and ZnO interface with the n-n heterojunction can accelerate electron transport. This is beneficial to the photocatalytic oxidation reaction of TC [7], facilitating its breakdown into smaller, innocuous molecules. Notably, with a VB potential of +2.91 V vs. NHE, ZnO exhibits more positive potential compared to the redox potential of $\cdot OH∕H_{2}O$ (+2.40 V vs. NHE) [36]. Consequently, the photogenerated holes react with $H_{2}O$ to generate hydroxyl radicals, which in turn catalytically oxidize TC into carbon dioxide and water. In addition, the effective holes present in the valence band of ZnO can directly contribute to the degradation of TC as an active species. In summary, these mechanisms underscore the pivotal roles played by oxygen superoxide anions, photogenerated electrons, and hydroxyl radicals in the efficient photocatalytic degradation of TC by TiO_2_-ZnO under simulated solar radiation conditions.

## 4. Conclusions

In this work, an n-n-type heterojunction TiO_2_-ZnO photocatalyst (50 mg of TiO_2_) was designed for the high degradation (97%) of tetracycline under visible light, which is 1.2 times higher than that of pure zinc oxide. This is achieved by constructing an n-n-type heterojunction between ZnO and TiO_2_ to form a built-in electric field, which promotes the separation of electron–hole pairs in zinc oxide. By incorporating anatase TiO_2_ into ZnO, creating an oxygen vacancy state, the performance is significantly enhanced by facilitating the separation and migration efficiency of the photogenerated charge carriers. Our results indicate that the n-n-type heterojunction TiO_2_-ZnO photocatalyst achieves high efficiency degradation of tetracycline under visible light, surpassing conventional traditional composite ZnO-based photocatalysts.

## Figures and Tables

**Figure 1 nanomaterials-14-01802-f001:**

A schematic illustration for the fabrication procedure of the TiO_2_-ZnO photocatalyst.

**Figure 2 nanomaterials-14-01802-f002:**
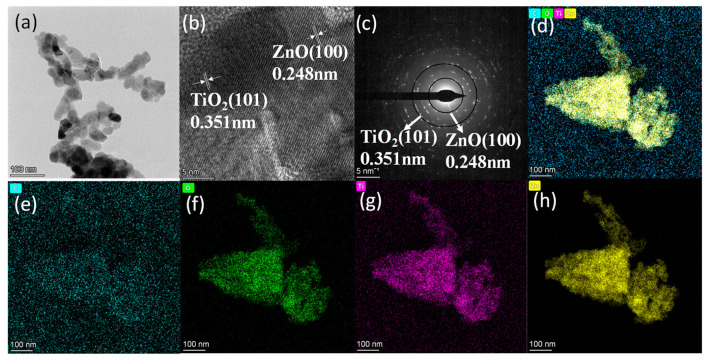
TEM (**a**), HRTEM (**b**,**c**), and EDS mapping (**d**–**h**) images of TZ samples.

**Figure 3 nanomaterials-14-01802-f003:**
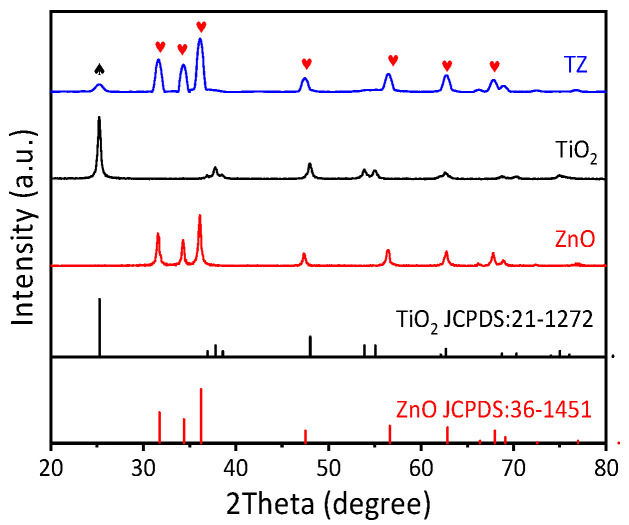
XRD patterns of ZnO and TZ.

**Figure 4 nanomaterials-14-01802-f004:**
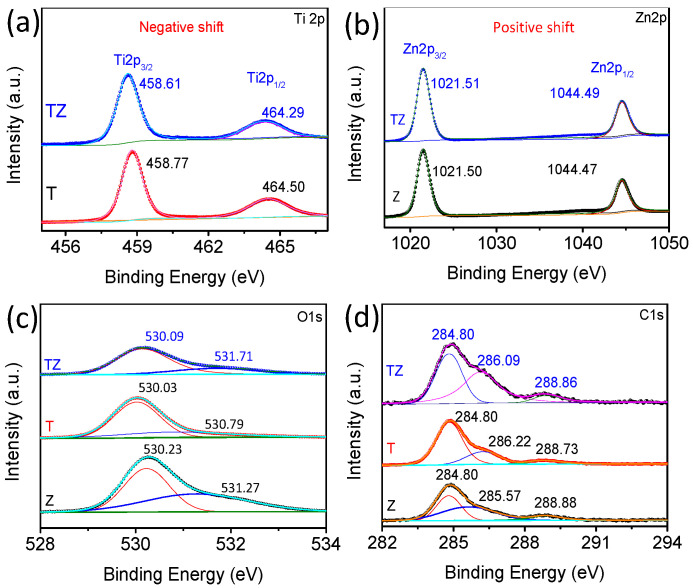
XPS spectra of ZnO, TiO_2_, and TZ: (**a**) Ti2p; (**b**) Zn2p; (**c**) O1s; (**d**) C1s.

**Figure 5 nanomaterials-14-01802-f005:**
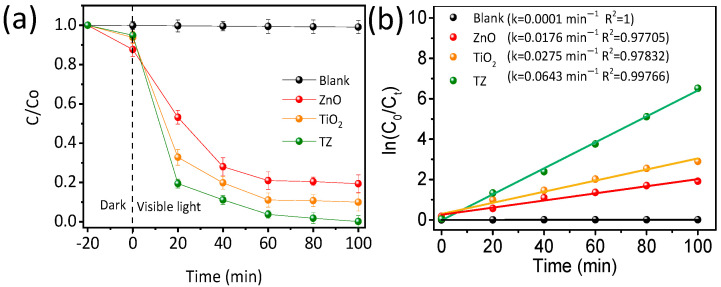
Photodegradation curves (**a**) and kinetic curves (**b**) of degrading TC under visible light.

**Figure 6 nanomaterials-14-01802-f006:**
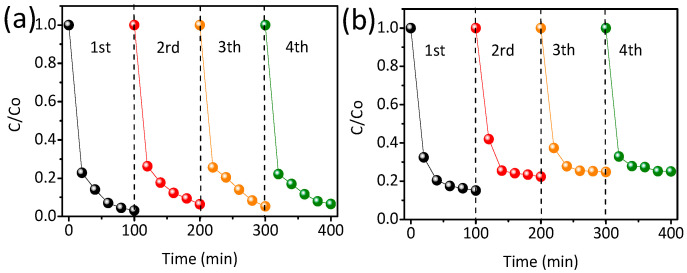
Recycling tests of TiO_2_-ZnO (**a**) and ZnO (**b**) for degrading TC.

**Figure 7 nanomaterials-14-01802-f007:**
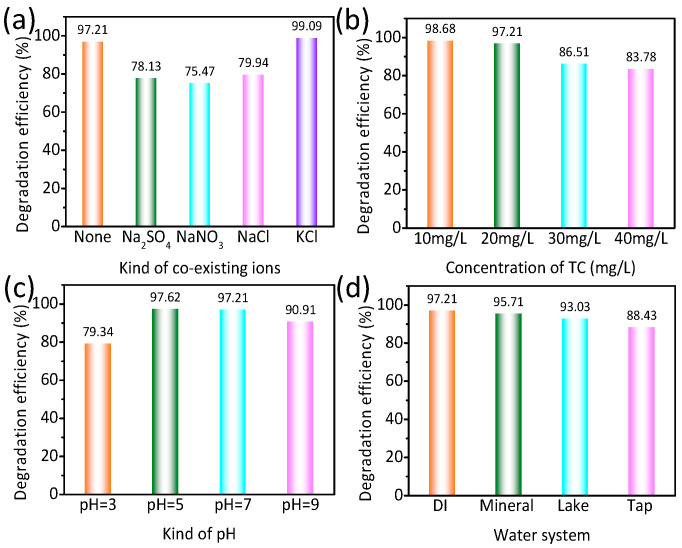
The influences of inorganic ions (**a**), concentration (**b**), pH value (**c**), and water source (**d**) on the degradation of TC by TiO_2_-ZnO under simulated sunlight.

**Figure 8 nanomaterials-14-01802-f008:**
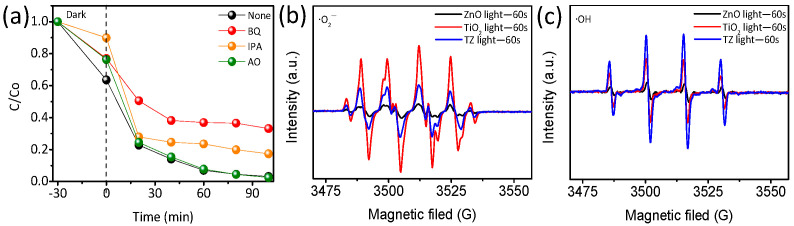
(**a**) Active species shown in ESR spectra of superoxide (**b**) and hydroxyl (**c**) radicals.

**Figure 9 nanomaterials-14-01802-f009:**
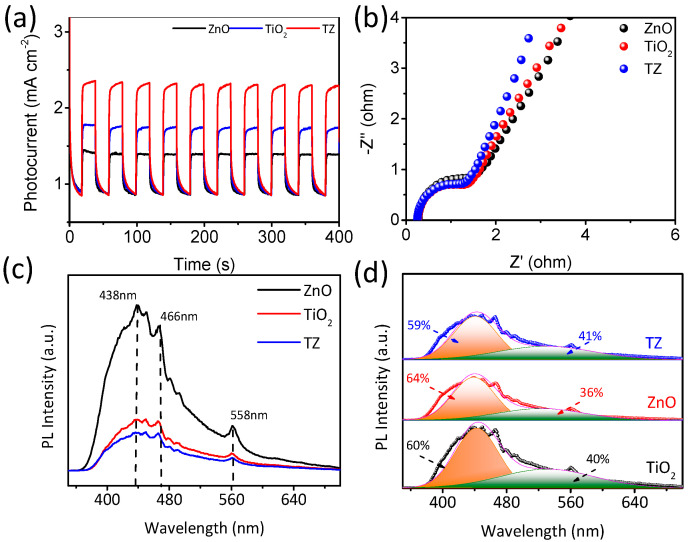
(**a**) Photocurrent responses; (**b**) EIS; (**c**) PL spectra; (**d**) PL spectra.

**Figure 10 nanomaterials-14-01802-f010:**
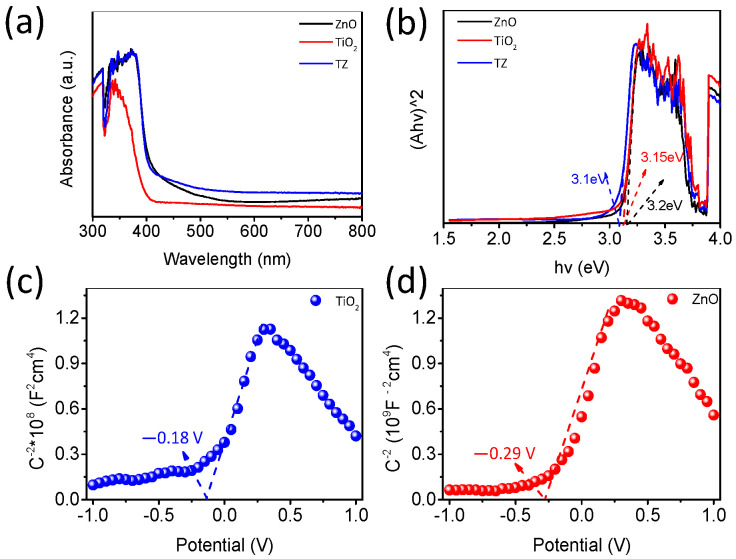
(**a**) UV–vis DRS spectra; (**b**) Tauck’s plots of (αhv)_2_ vs. hv; (**c**,**d**) MS.

**Figure 11 nanomaterials-14-01802-f011:**
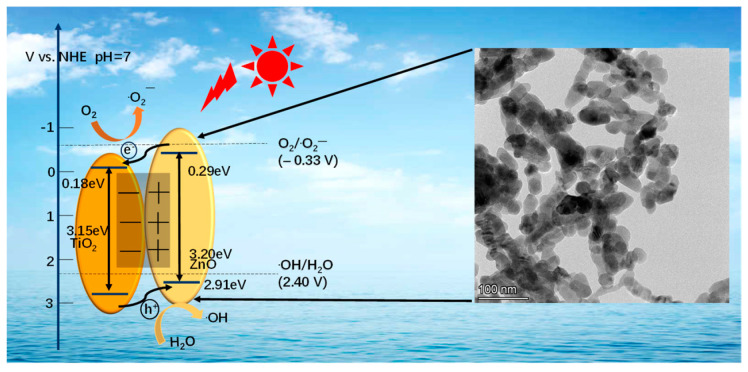
Proposed mechanism for photocatalytic degradation of TC by TiO_2_-ZnO.

**Table 1 nanomaterials-14-01802-t001:** Photodegradation efficiency of different forms of modified ZnO under visible light for 1 h.

Photocatalysts	Catalyst (mg/mL)	TC (mg/L)	Photodegradation Efficiency	Refs.
TiO_2_-ZnS@ZnO	0.3	20	87%	[9]
AgIn_5_S_8_/ZnO	0.5	10	88%	[11]
LaCoO_3/_ZnO	0.2	20	80%	[28]
CDots/Ag/ZnO	0.3	30	96%	[32]
ZnO/CuS	0.4	30	75%	[33]
TiO_2_-ZnO	0.2	20	97%	Our work

## Data Availability

Data are contained within the article.

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
