# Peer review of "Synthesis of TiO2-ZnO n-n Heterojunction with Excellent Visible Light-Driven Photodegradation of Tetracycline"

_nanomaterials, 2024, doi:10.3390/nano14221802_

Round 1

Reviewer 1 Report

Comments and Suggestions for Authors

This work deals with the preparation of TiO2-ZnO catalysts for the removal of tetracycline by visible light photodegradation. This is an interesting work with a deep study of this kind of catalysts and good results and I recommend its publication in Nanomaterials after a few points are addressed:

-       Lines 86-87: “and stirred 86 for 20 min in darkness to reach the adsorption-desorption equilibrium”. Was it checked that this time of 20 min was enough to reach the adsorption-desorption equilibrium?

-       Line 95: “Error! Reference source not found.” Remove this sentence and add “Figure 1” in its place.

-       The same as the previous point with the corresponding figures in lines 104, 106, 108, 121, 136, 138, 148, 154, 160, 178, 193, 202-204, 216, 221, 228, 239, 255, 263, 284, 293, 297, 308, 345, etc..

-       Line 119: revise the spaces between letters and parenthesis: “TEM(a)” must be replaced by “TEM (a)” and more.

-       Figure 3: the inner legends with the TiO2 and ZnO JCPDS information are tough to see in the place where they are now, better move them to the upper part of the figure. Moreover, the Y axis is not well selected since the XRD pattern placed at the bottom of the figure has no background line.

-       Figure 5b: the fitting of the experimental data to a pseudo-fist-order kinetics is not very good and the regression coefficients (R2) have not been included anywhere, neither in the figure nor in any table. Why has only the pseudo-first-order kinetics been tested? Why not a pseudo-second-order? The authors should test some different reaction orders, obtain their corresponding regression coefficients and, then, choose the better fitting.

-       Line 248: revise the spaces between letters and parenthesis: “ions(a)” must be replaced by “ions (a)” and more.

Author Response

Reviewer: This work deals with the preparation of TiO2-ZnO catalysts for the removal of tetracycline by visible light photodegradation. This is an interesting work with a deep study of this kind of catalysts and good results and I recommend its publication in Nanomaterials after a few points are addressed:

  1. Lines 86-87: and stirred for 20 min in darkness to reach the adsorption-desorption equilibrium. Was it checked that this time of 20 min was enough to reach the adsorption-desorption equilibrium?

Answer 1 (A1): Through adsorption experiments, it was observed that the adsorption reaction reached equilibrium after a minimum of 20 minutes, and the adsorption remained stable thereafter. (Please refer to Fig. S7 in the supporting information)

  1. Line 95: Error! Reference source not found. Remove this sentence and add Figure 1 in its place.

Answer 2 (A2): “Error! Reference source not found.” Remove this sentence and add Fig. 1 in its place. (Please refer to page 3 on line 95 in the manuscript)

  1. The same as the previous point with the corresponding figures in lines 104, 106, 108, 121, 136, 138, 148, 154, 160, 178, 193, 202-204, 216, 221, 228, 239, 255, 263, 284, 293, 297, 308, 345, etc.

Answer 3 (A3): “Error! Reference source not found.” Remove this sentence and add related Fig. X (X= 2,3,4,5,6,7,8,9,10,11) in its place. (Please refer to manuscript)

  1. Line 119: revise the spaces between letters and parenthesis: TEM(a) must be replaced by TEM (a) and more.

Answer 4 (A4): Thanks for your careful work. We have carefully checked the manuscript to revise the spaces between letters and parenthesis and more. (Please refer to page 3 on line 118 and more in the manuscript)

  1. Figure 3: the inner legends with the TiO2 and ZnO JCPDS information are tough to see in the place where they are now, better move them to the upper part of the figure. Moreover, the Y axis is not well selected since the XRD pattern placed at the bottom of the figure has no background line.

Answer 5 (A5): Thanks for your careful work. We have carefully checked the manuscript to revise the Fig. 3. (Please refer to page 4 on line 130 and more in the manuscript)

  1. Figure 5b: the fitting of the experimental data to a pseudo-fist-order kinetics is not very good and the regression coefficients (R2) have not been included anywhere, neither in the figure nor in any table. Why has only the pseudo-first-order kinetics been tested? Why not a pseudo-second-order? The authors should test some different reaction orders, obtain their corresponding regression coefficients and, then, choose the better fitting.
  • Answer 6 (A6): Thanks for your careful work. We are sorry for not classifying this point clearly. As described in Fig. 5a of our manuscript, we have added error bars to the fitting of the experimental data. Considering that no species can ensure a 100% efficiency to the pseudo-first-order, we therefore employed it as the pseudo-first-order. According to you’re your suggestion, we have now performed regression coefficients (R2) in Fig. 5b. (Please refer to page 6 on line 194 and more by using yellow for annotation in the manuscript)
  1. Line 248: revise the spaces between letters and parenthesis: “ions(a)” must be replaced by “ions (a)” and more.

Answer 7 (A7): Thanks for your careful work. We have carefully checked the manuscript to revise the spaces between letters and parenthesis and more. (Please refer to page 8 on line 241 and more by using yellow for annotation in the manuscript)

Reviewer 2 Report

Comments and Suggestions for Authors

The manuscript is well-written and the results are presented in a comprehensive manner. Overall the work is good. However, further modifications are still needed before its acceptance. Some specific comments are listed below.

1.      The author states the novelty of the work in the introduction section.

2.      Sec. 2.1, synthesis part, the author should include the purity of the precursors.

3.      The figure caption number was not shown in the entire manuscript. The author should rectify it.

4.      Fig.S3, Most of the peaks are not labelled. At ~500 a peak appeared sharply. What is the peak of the element and why does the author not mention it?

5.      Fig.S4, the author should include the statistical analysis to confirm the differences in the photocatalytic performance. The values are too similar.

6.      The author did not explain Fig. S4 in the main body of the text.

7.      The author should include the numerical results in the conclusion section.

Author Response

Please resubmit a  revised version of the graphical abstract.

 We have carefully cheched the manuscript to revise the graphical abstract.
